# Multilocus sequence typing database for *Streptococcus agalactiae* contains a spurious allele of the transketolase gene

Swaine L. Chen,[1,2] Suma Tiruvayipati,[1] Wen Ying Tang,[3] Timothy M. S. Barkham[3]

**ABSTRACT** The *tkt* (transketolase) gene is one of the seven gene fragments used in the multilocus sequence typing (MLST) system for *Streptococcus agalactiae*. We discovered that the tkt_134 allele is derived from a homologous gene (which we designate *tktX*) that is not present in all *S. agalactiae*; all known strains that contain a match to the tkt_134 allele also contain a gene sequence that is much closer in sequence identity to the other non-tkt_134 alleles (i.e., the canonical tkt gene) in the database. Based on these data, the tkt_134 allele has been removed from the MLST database as of September 2021, and all sequence types containing tkt_134 have also been removed.

**IMPORTANCE** Multilocus sequence typing (MLST) databases are a common good and remain important for research, medical, and epidemiological purposes. This remains true even in the context of widespread whole-genome sequencing. We discovered a contaminating allele of the *tkt* gene in the *S. agalactiae* MLST database that led to unstable, ambiguous, or erroneous MLST assignment. The allele has since been removed from the public database based on the results presented in this manuscript.

**KEYWORDS** MLST, GBS, *S. agalactiae*, molecular epidemiology

Multilocus sequence typing (MLST) is a widespread, convenient, and powerful DNA sequence-based typing system that has been applied to numerous bacteria (1). Particularly for medically important pathogens, MLST provides a rapid way to reliably assign coherent subspecies groups that are more stable across time and across different laboratories than previous alternatives (2). As such, it has contributed greatly to the identification, monitoring, and correspondence of bacterial outbreaks (2–5). Due to this combination of convenience and precision, MLST continues to be useful today even as genomics becomes more economical and prevalent (6, 7). In fact, despite the undisputedly higher resolution of whole-genome sequencing (WGS), MLST remains a commonly reported output from bacterial WGS projects.

MLST systems are typically defined using six to seven "housekeeping genes." To take the specific example of *Streptococcus agalactiae* (also known as group B streptococcus, GBS), typing a given strain of GBS requires obtaining the sequence of fragments of the following seven genes: *adhP*, *pheS*, *atr*, *glnA*, *sdhA*, *glcK*, and *tkt* (8). Each of these seven sequences is assigned an allele number based on matching its sequence to a database of known alleles. If the sequence of one of these genes is not present in the existing database, a new allele is assigned. With the seven allele numbers, an MLST profile can be constructed, and a single sequence type (ST) refers to a specific combination of seven numbers. An MLST system thus must balance two opposing considerations: (i) the genes used for typing should be, to the extent possible, conserved and present in one and only one copy in all individuals of the species (if one of the genes is missing, or two alleles of a gene are present, no ST can be assigned), and (ii) the genes used should have enough genetic diversity to provide meaningful discrimination between strains.

Address correspondence to Swaine L. Chen, swainechen@gmail.com.

The authors declare no conflict of interest.

See the funding table on p. 8.

We discovered that the *tkt* allele database for the *S. agalactiae* MLST system contained a divergent allele, tkt_134. The tkt_134 sequence corresponds to a gene that is homologous but clearly genetically distinct from the gene represented by the other *tkt* alleles in the database (and thus likely a paralog), and all strains we identified carrying the tkt_134 sequence also contain a sequence for the canonical *tkt* gene. Based on observations reported here, the tkt_134 sequence has been removed from the MLST database at http://pubmlst.org as of August 2021, and associated STs have also been removed. For clarity, we henceforth will refer to the specific DNA sequence in the MLST database as tkt_134, the paralogous gene giving rise to allele tkt_134 as *tktX*, and the gene represented by all other MLST alleles as of September 2021 as the "canonical" *tkt* or simply *tkt*.

## RESULTS

### The tkt_134 allele is homologous to, but genetically divergent from, all other *tkt* alleles

During a whole-genome sequencing project for GBS strains, one of our standard analyses was to predict MLSTs using the unassembled short reads directly (via SRST2). All MLST predictions were further validated by using BLASTN with the MLST databases on the assembled sequences. The reference MLST alleles were downloaded from the http://pubmlst.org website in August 2021. Those strains that did not exactly match an existing ST were submitted to the PubMLST website for automated MLST assignment. Four strains (SG-M23, SG-M635, SG-M803, and SG-M636) were predicted to have two *tkt* alleles, with the tkt_134 allele being common to all, and thus no ST was assigned. A search on the PubMLST website for other strains with the tkt_134 allele showed two existing STs (ST1693 and ST1707) with only tkt_134 and one additional strain (GU1559) with two *tkt* alleles (and thus no ST assignment).

We created a phylogenetic tree of all the existing *tkt* alleles in the PubMLST database (Fig. 1). The tkt_134 allele was clearly an outlier in terms of genetic diversity. Excluding tkt_134, all other alleles had an average sequence identity >99% to the other *tkt* alleles. The tkt_134 allele, in contrast, was on average only 75% identical to the other alleles.

### The tkt_134 allele represents a distinct, but homologous, gene (*tktX*)

Given the divergence of tkt_134 and the predicted presence of two alleles in some of the strains, we hypothesized that tkt_134 might have been derived from a distinct gene from the canonical *tkt* found in most GBS. We therefore examined the genomes of three commonly studied and fully sequenced GBS strains, A909 (9), SG-M1 (10), and COH1 (9, 11). Interestingly, we found both the expected tkt_2 allele (based on its entry in the PubMLST database as an ST7 strain) as well as a tkt_134 allele in A909, in different genomic contexts. Specifically, the canonical *tkt* is flanked by SAK_RS01730 (encoding a helix-turn-helix protein) and SAK_RS01740 (encoding a bacteriocin immunity protein), while *tktX* is flanked by the SAK_RS08825 (a gene encoding a DUF 4365 domain protein) and SAK_RS08835 (fsa, fructose-6-phosphate aldolase) genes. Similarly, SG-M1 had sequences that corresponded to both tkt_2 and tkt_134, with the same flanking genes (>99% sequence identity for all genes examined). COH1 had the expected tkt_1 allele in the same genomic context as the other canonical *tkt* alleles in the other strains, but no convincing sequences match (blastn reported no hits at an *e*-value cutoff of 10) to *tktX* or the immediately flanking genes (Fig. 2A).

We then asked whether the inclusion of tkt_134 in the database might be due to misidentifying the tkt_134 allele (from the *tktX* locus) as a *bona fide tkt* gene allele. In particular, could issues with genomic analysis lead to not identifying the canonical *tkt* allele? To examine this, we downloaded the published assembly for CCUG 28551 (the ST1691 strain reported to only have the tkt_134 allele). Using blast against the A909 reference sequence, we indeed saw that the region used for typing MLST in the canonical *tkt* locus was not present in CCUG 28551 (Fig. 2B).

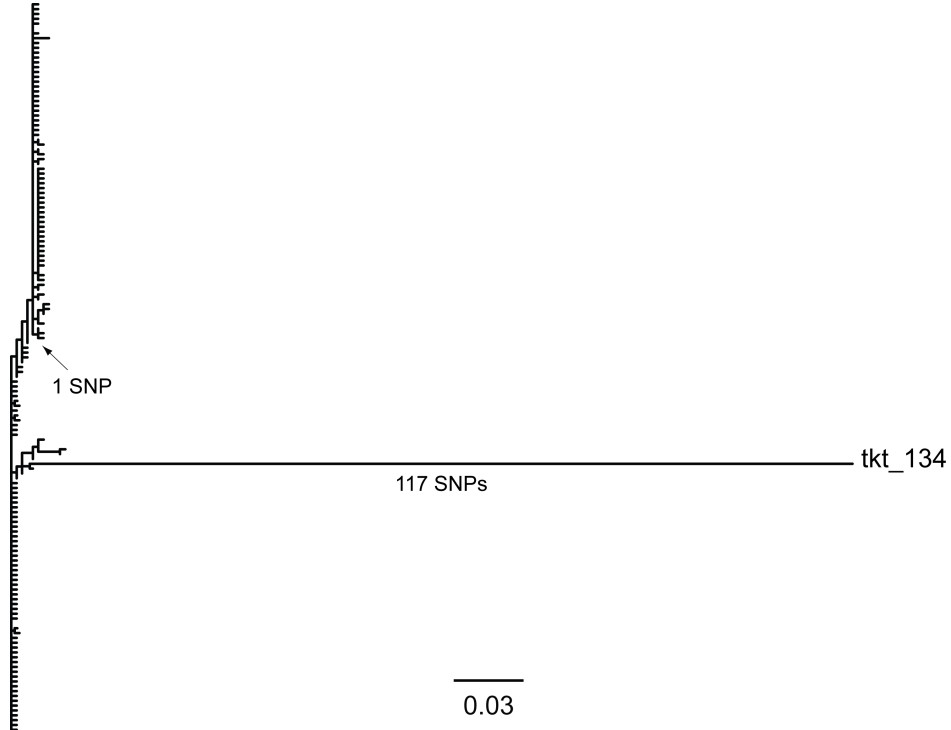

**FIG 1** Unrooted approximately maximum likelihood phylogenetic tree of the 151 *tkt* alleles in the PubMLST database as of August 2021. The alignment is over 480 nt with no gaps. The scale bar is indicated at the bottom; furthermore, a branch representing a single SNP difference is indicated by the black arrow. The tip corresponding to tkt_134 is labeled, and the number of SNPs in the branch leading to tkt_134 is indicated.

To explore the hypothesis that the canonical *tkt* gene actually does exist in this strain but may have been missed initially, we then downloaded the raw Illumina reads for this strain (SRR494268). We assembled this using the same pipeline as the four strains we had recently sequenced and repeated the blast analysis, discovering that both the canonical *tkt* and the *tktX* regions were present in the assembly, and the *tkt* allele corresponded to tkt_2, while the *tktX* sequence corresponded to tkt_134. We further checked whether sequence coverage or the homology between *tkt* and *tktX* could have been an issue during the original assembly or indicate problems with our new assembly. As shown in the coverage plots in Fig. 2B, we saw no evidence of mismapping or abnormal coverage across the entire *tkt* and *tktX* genes, suggesting that these sequences were assembled normally by modern assemblers (as would be expected for sequences that are only 75% identical). A similar analysis of the other four SG-M strains we sequenced and assembled gave similar results, that all had both a canonical *tkt* locus and a *tktX* locus, and the tkt_134 allele in each strain corresponded to the *tktX* locus.

## The presence of *tktX* does not interfere with PCR-based MLST typing of the existing *tkt* gene

We asked whether the presence, albeit variable, of the paralogous *tktX* gene in some GBS strains might suggest that the existing scheme should be refined or redefined. We used the recommended PCR primers from the PubMLST website to perform PCR on genomic DNA isolated from six strains. Four of the strains, based on genome sequencing, contained both a canonical *tkt* and a *tktX* allele. Two of the strains contained only a canonical *tkt* by genome sequencing and were chosen as controls (that were originally isolated and sequenced contemporaneously with the other four strains). PCR for the *tkt* gene revealed a single band in all strains (Fig. 3), and Sanger sequencing confirmed that this PCR product was from the canonical *tkt* allele and matched the genome

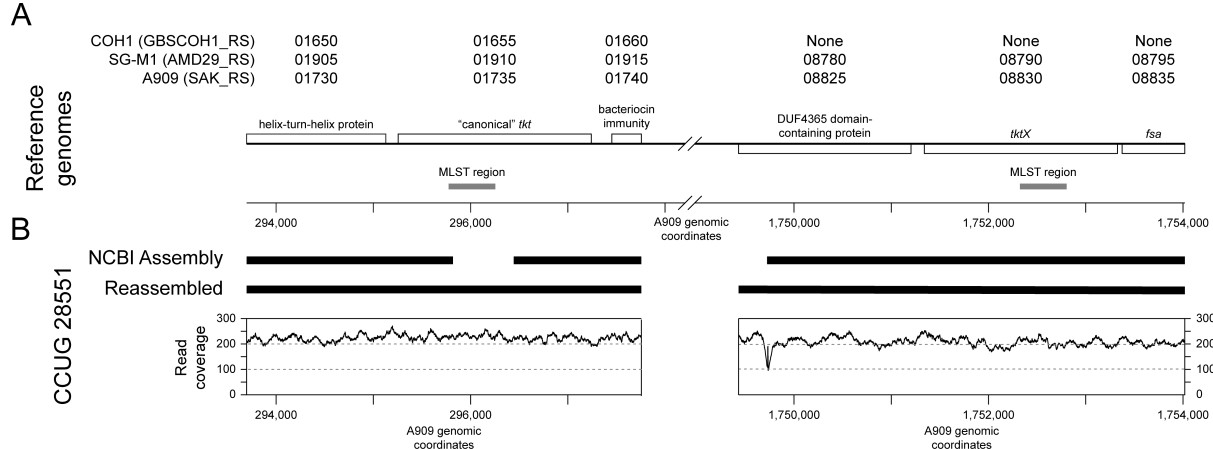

**FIG 2** Genome comparisons of the canonical *tkt* and the *tktX* loci. A reference genetic map, which is used in common in all parts of the figure, is shown using white boxes to represent genes; boxes above the line indicate the coding strand proceeds from left to right on this diagram, while boxes below indicate the opposite orientation. Common gene names/annotations are indicated above each white box. (A) Corresponding systematic gene identifiers (numbers only) are indicated for three reference genomes (prefixes are indicated in parentheses, i.e., the systematic name for the first A909 gene indicated is SAK_RS01730). Gene coordinates from the A909 genome are shown just below the genetic map for reference. Short gray bars indicate the regions that are homologous to the *tkt* gene region used for MLST. (B) CCUG 28551 genome analysis. Thick black bars indicate blastn matches to the corresponding A909 sequence [using the common genetic map and coordinates from (A)] for the published NCBI assembly and a *de novo* assembly generated from the raw Illumina reads ("Reassembled"). Coverage maps generated by aligning the raw Illumina reads to the A909 genome are shown at the bottom, again using the same genomic coordinates as in (A). All indications of sequence homology are >99% by blastn for individual genes in (A) as well as larger chromosomal segments in (B).

sequencing data for the *tkt* gene. Therefore, the presence of the *tktX* gene in a strain does not generally interfere with MLST by PCR and Sanger sequencing using the existing recommended primers.

## DISCUSSION

Based on our results, we conclude that tkt_134 represents a sequence of a gene homologous to (but not orthologous to) the canonical *tkt* gene intended to be used in the GBS MLST system; we refer to this homologous second gene as *tktX*. To summarize the evidence for this:

1. Tkt_134 has a distinctly divergent sequence compared with the other *tkt* alleles in the database as well as a distinct genomic context.
2. All GBS strains we examined carry a *tkt* gene, while only a subset of these additionally carries a *tktX* gene based on sequence context.
3. The two GBS strains in the MLST database that are reported only to have tkt_134 (indicative of *tktX*) appear also to have a canonical *tkt* allele that does not assemble well with the available genome sequence data.

With the further information that the presence of the *tktX* gene does not seem to interfere with typing using the current PCR primers for the *tkt* gene, we therefore made the suggestion to remove the tkt_134 gene from the MLST database as well as the two STs that have tkt_134 in their profile, which was performed as of September 2021.

Given that there was no ambiguity in using PCR to type *tkt* in strains that also carry the *tktX* gene, we suspect that the inclusion of the tkt_134 allele in the reference MLST database was due to the inability to find a canonical *tkt* allele in genome sequencing data for one or both of the strains forming the basis for the assignment of ST1693 and ST1707. Interestingly, the genomic region containing *tktX* was well assembled from the publicly available genome sequencing data for the strains registered in the MLST database for these two STs. We, therefore, further suggest that the initial inclusion of the *tktX* sequence in the MLST database was due to the use of WGS for MLST, allowing the

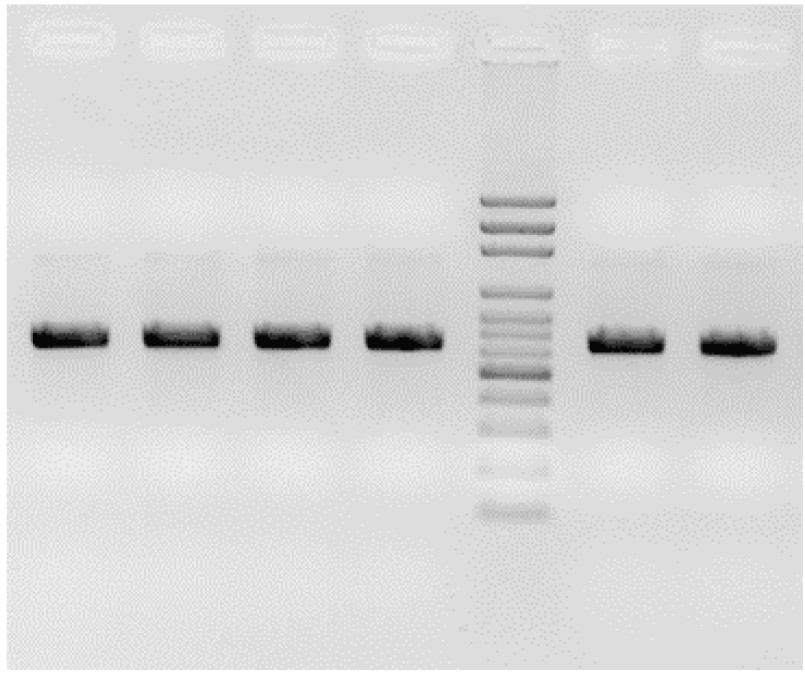

**FIG 3** PCR on genomic DNA using existing primers recommended for amplifying *tkt* yields a single band regardless of the presence of *tktX*. Lane 5 (counting from the left) is a 100-bp DNA size ladder, and other lanes are PCRs from the indicated strains. The four strains to the left of the ladder are predicted to carry *tktX*, while the two on the right are not.

detection of homologous sequences that are too divergent to be amplified by the MLST PCRs. While sequence quality and quantity are key determinants of whether a region can be assembled, other technical reasons such as DNA quality, quantity, and purity (e.g., contamination with another GBS or another organism) play a role as well. We did not explore any of these possibilities, both due to a lack of data and because this would not alter our conclusions. However, these results suggest a valuable role for considering including sequence divergence cutoffs or PCR verification of new alleles identified solely through genomic MLST analysis.

The fact that, at some point, the tkt_134 allele was included in the MLST database raises the question of whether *tkt* is a "good" housekeeping gene to use for the MLST system for GBS. By the criteria of universal presence in single copy in all GBS strains, it does seem like a good gene (while we can never be certain of universal presence, we found no conclusive evidence of its absence in any strain). Assuming that the root cause of the initial inclusion of tkt_134 in the database was due to some technical reason whereby sequence data for *tkt* were lacking, potentially due to issues with assembly that have been resolved in the past 10 years, the presence of a reasonably close paralog may raise concerns, particularly for strains where older assemblies but not raw sequencing data are available. However, we note that many labs have had no problem distinguishing between *tkt* and *tktX*. In addition, A909, one of the first GBS genomes sequenced (12), carries both genes and was assigned an MLST when it was sequenced (9); perhaps most importantly, typing by PCR + Sanger sequencing using the existing primers gives a single (correct) *tkt* allele in strains that do and do not carry *tktX*. We, therefore, do not see any clear need to redefine the MLST scheme for GBS to use a different gene from *tkt*.

Why not just let the tkt_134 allele remain in the database? There is a possibility that it could recombine into the canonical *tkt* locus. However, inclusion of *tktX* alleles in the database does degrade the MLST scheme; as an anecdotal example, we noticed that the *tkt* allele assignment was unstable for strains carrying both *tkt* and *tktX*, and this stability disappeared when the *tktX* allele was removed from the reference database. [It is of interest to note that *tkt* alleles from strains like A909 that do carry a *tktX* gene were largely added to the database (without issue) prior to the addition of tkt_134, and we now see instability in the WGS-based MLST assignment for these well-known strains when we use databases that contain tkt_134.] While GBS is known to be recombinogenic on evolutionary time scales, the population structure is marked by distinct and relatively stable clones (11, 12); thus, it is quite likely that *tkt* has been the target of some recombination at some point in some subset of GBS, though until now we see no evidence for its replacement by divergent homologs in any GBS. Thus, the balance points toward removing tkt_134.

Recombination may also be directly relevant to the presence of *tktX* in only a subset of GBS strains. The relative clonality of GBS has facilitated the identification of large-scale recombination between different clones of GBS. Salient examples [among many; see reference (12) for a more complete discussion and additional references) include cases of capsular serotype switching (13), variable presence of genomic islands (including conjugative and integrative elements, which frequently transfer resistance genes) (11), and genome reduction (14). The fact that *tktX* and flanking genes are not present in many GBS strains could be consistent with either loss (from an ancestral state where *tktX* was conserved in all GBS) or gain through recombination. We found the same genomic context (at least for the immediate flanking genes) with 99.91%–99.98% identity in ST1 (SG-M23), ST2 (CCUG 28551), ST7 (A909), and ST283 [SG-M1, SG-M635, SG-M636, and SG-M803 (an ST1863 strain, a single-locus variant of ST283)] strains, while the rest of the genomes are as low as 99.5% identical (data not shown), which suggests that the *tktX* locus may indeed be spreading among GBS strains by recombination. Thus, we speculate that *tktX* may have originally been acquired through recombination by one or a few clones of GBS. Furthermore, because the genomic context for *tktX* is distinct from that for *tkt*, we reason that *tktX* likely did not arise by gene duplication within GBS. Thus, we finally speculate that the ultimate origin of *tktX* may be from another organism and that, like many other genes, it was introduced into one or a few clones of GBS at some point in the past. However, all of these speculations are based on limited data and bear further investigation. For the purposes of this manuscript, the results are clear that *tktX* is indeed distinct from *tkt* and should be removed from the MLST database.

The removal of tkt_134 leads to the removal of both ST1693 and ST1707 (and reassignment of any representatives to other STs). We questioned whether this might alter any epidemiological conclusions. A search on PubMed showed no results for either ST1693 or ST1707, leading us to suspect that this change will not disrupt previously published results, though we did not pursue this further.

This *tktX* example suggests that outliers in genetic diversity in the existing allele databases may be a generally useful indicator of some quality issues. If additional sequence information (such as WGS) or additional PCR primers targeted to flanking regions are available, these would be useful supporting data to determine whether a divergent sequence result for an MLST gene is indeed a *bona fide* new allele. The alternative likely points to other technical issues, such as in the strain isolation, DNA purification, sample handling, sequencing, or assembly processes.

We do expect that technical issues related to sequencing, assembly, and other downstream analysis will continue to improve. Lab-based issues related to sample purity and DNA quality are likely harder to uniformly improve, especially given that laboratory facilities and training can vary with geography and resources. This is further amplified by the fact that understanding the diversity and molecular epidemiology of GBS requires more sampling from traditionally undersampled geographical areas, which highly correlates with lower resources. All of these support the enduring importance

of MLST as a rapid, economical, and reliably comparable typing system for GBS and other bacteria. Maintaining these databases, both by addition of strains and correction of errors like the one reported here, will thus remain an important activity for the community for the foreseeable future.

## MATERIALS AND METHODS

### Phylogenetic analysis of *tkt* alleles

We downloaded the fasta sequences for all *tkt* alleles in the MLST database in August 2021; this database included 151 *tkt* sequences, including the tkt_134 allele, which we refer to as the *tktX* allele in this manuscript. These sequences were all 480 nt and already aligned. We created an unrooted approximately maximum-likelihood phylogenetic tree using FastTree 2.1.8 (15) with the –gtr and –nt command line options. This phylogenetic tree was visualized with GGTREE 3.8 (16) in R (4.3.3) (https://www.R-project.org).

### Genome sequences

All genome sequences described were already available on Genbank; accession numbers and references are shown in Table 1 as well as whether *tktX* was detected by blastn in that strain.

### WGS-based MLST

MLST predictions were made using SRST2 0.2.0 (18) on the unassembled reads. Reads were also assembled both using the GIS GERMS Platform [using velvet 1.2.10 (19), OPERA 1.4.1 (20), and FinIS 0.3 (21)] and using SKESA 2.5.1, then MLST was called from the assemblies using a custom BLASTN (22)-based script for fully assembled reference sequences; the results were the same regardless of assembler. In all cases, the GBS MLST databases were downloaded from http://pubmlst.org/sagalactiae/. Data for coverage plots were generated by mapping the paired reads in FASTQ format for each data set to the A909 reference genome using the bwa mem algorithm (version 0.7.17) (23) with default settings and the SAMtools utilities (version 1.13) (24) to extract coverage depth.

### PCR of the *tkt* gene fragment used for MLST

GBS isolates were subcultured onto blood agar, and isolates were reconfirmed to be GBS using a MALDI-TOF system (Bruker). Genomic DNA was extracted with the QIAamp DNA Mini Kit (Qiagen), then quantified and quality checked with the Nanodrop spectrophotometer ND1000 and 1% agarose gel electrophoresis. The *tkt* gene was amplified using the PCR primers recommended on the PubMLST site (https://pubmlst.org/organisms/streptococcus-agalactiae/primers): forward primer (5′-CCAGGCTTTGATTTAGT

**TABLE 1** Details of strains and sequencing data sets

| Strain | Assembly accession | Short-read accession | MLST | tktX detected | Reference |
|---|---|---|---|---|---|
| A909 | GCF_000012705.1 | | ST7 | Y | (9) |
| SG-M1 | GCF_001275545.2 | | ST283 | Y | (10) |
| COH1 | GCF_000689235.1 | | ST17 | N | (9, 11) |
| CCUG 28551 | GCF_000310585.1 | SRR494268 | ST2 (ST1691 prior to this work) | Y | BioProject PRJNA214546; no results in Pubmed |
| SG-M23 | | SRR15143442 | ST1 | Y | (17) |
| SG-M340 | | SRR15143445 | ST335 | N | (17) |
| SG-M635 | | SRR15143445 | ST283 | Y | (17) |
| SG-M636 | | SRR15143442 | ST283 | Y | (17) |
| SG-M803 | | SRR8052387 | ST1863 (SLV[a] of ST283) | Y | (17) |
| SG-M811 | | SRR15143884 | ST17 | N | (17) |

[a]SLV, single-locus variant.

TGA-3′) and reverse primer 5′-AATAGCTTGTTGGCTTGAAA-3′). We used the HotStar Taq Master Mix Kit (Qiagen) [15 µL HotStar Taq Buffer, 1 µL of each primer (10 µM), 5 µL diluted DNA, in a 30 µL reaction). Cycling conditions were 95℃ for 15 min and then 35 cycles of 94℃ for 30 s, 56℃ for 60 s, 72℃ for 60 s, and a final elongation of 3 min at 72℃. Amplicons were purified with QIAquick PCR Purification Kit (Qiagen) prior to electrophoresis on a 1% agarose gel and Sanger sequencing (Axil Scientific, Singapore).

## ACKNOWLEDGMENTS

The authors thank Odile Harrison for helpful discussion, for facilitating removal of the tkt_134 allele from the MLST database, and for assisting to maintain the GBS MLST database in general. The authors further acknowledge all involved in maintaining the MLST databases and keeping them as a useful public resource.

This work was supported in part by the National Medical Research Council, Ministry of Health, Singapore (grants NMRC/CIRG/1467/2017 and CIRG19NOV-0024); the Temasek Foundation Innovates through its Singapore Millennium Foundation Research Grant Programme, and the Genome Institute of Singapore (GIS)/Agency for Science, Technology and Research (A*STAR).

## AUTHOR AFFILIATIONS

[1]Infectious Diseases Translational Research Programme, Department of Medicine, Division of Infectious Diseases, Yong Loo Lin School of Medicine, National University of Singapore, Singapore, Singapore
[2]Laboratory of Bacterial Genomics, Genome Institute of Singapore, Singapore, Singapore
[3]Department of Laboratory Medicine, Tan Tock Seng Hospital, Singapore, Singapore

## PRESENT ADDRESS

Swaine L. Chen, New York, New York, USA

## AUTHOR ORCIDs

Swaine L. Chen  http://orcid.org/0000-0002-0107-2861
Timothy M. S. Barkham  http://orcid.org/0000-0003-0975-2244

## FUNDING

| Funder | Grant(s) | Author(s) |
| --- | --- | --- |
| MOH \| National Medical Research Council (NMRC) | NMRC/CIRG/1467/2017 | Swaine L. Chen |
| | | Suma Tiruvayipati |
| MOH \| National Medical Research Council (NMRC) | CIRG19NOV-0024 | Swaine L. Chen |
| | | Suma Tiruvayipati |
| Temasek Foundation (Temasek Foundation International) | Singapore Millennium Foundation Research Grant Programme | Swaine L. Chen |
| | | Wen Ying Tang |
| | | Timothy M. S. Barkham |
| Genome Institute of Singapore (GIS) | | Swaine L. Chen |

## AUTHOR CONTRIBUTIONS

Swaine L. Chen, Conceptualization, Data curation, Formal analysis, Funding acquisition, Investigation, Methodology, Project administration, Resources, Supervision, Validation, Visualization, Writing – original draft, Writing – review and editing | Suma Tiruvayipati, Data curation, Investigation, Methodology, Writing – review and editing | Wen Ying Tang, Formal analysis, Investigation, Methodology, Validation | Timothy M. S. Barkham, Data

curation, Formal analysis, Investigation, Project administration, Resources, Supervision, Writing – review and editing

## ADDITIONAL FILES

The following material is available online.

### Open Peer Review

**PEER REVIEW HISTORY (review-history.pdf).** An accounting of the reviewer comments and feedback.

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
