## [Reviewer comments · Microbiology Spectrum]

Microbiology Spectrum

Multilocus sequence typing database for *Streptococcus agalactiae* contains a spurious allele of the transketolase gene

Swaine Chen, Suma Tiruvayipati, Wen Ying Tang, and Timothy Mark Sebastian Barkham

Corresponding Author(s): Swaine Chen, National University of Singapore

Review Timeline:

Submission Date:	February 27, 2024
Editorial Decision:	April 13, 2024
Revision Received:	April 22, 2024
Accepted:	May 30, 2024

Editor: Sadjia Bekal

Reviewer(s): Disclosure of reviewer identity is with reference to reviewer comments included in decision letter(s). The following individuals involved in review of your submission have agreed to reveal their identity: Laura Maria Andrade de Oliveira (Reviewer #1); David MacMillan (Reviewer #2)

Transaction Report:

DOI: <https://doi.org/10.1128/spectrum.00537-24>

Re: Spectrum00537-24 (Multilocus sequence typing database for *Streptococcus agalactiae* contains a spurious allele of the transketolase gene)

Dear Dr. Swaine L. Chen:

Thank you for the privilege of reviewing your work. Below you will find my comments, instructions from the Spectrum editorial office, and the reviewer comments.

Revision Guidelines

Sincerely,
Sadjia Bekal
Editor
Microbiology Spectrum

Reviewer #1 (Comments for the Author):

In this article, the authors describe a tkt allele (tkt_134) of a homologous gene (tktX) that is not present in all *S. agalactiae* strains and can lead to r erroneous MLST assignment. This is an interesting and well written research article with clear goals, study parameters and procedures. I have just one comment to be considered by the authors for revision.

It is not clear in the methods which genomic data analysis were conducted to identify and compare the tkt_134 and tktX sequences and the canonical tkt sequence. The authors should describe these analyses more clearly in the text.

Reviewer #2 (Comments for the Author):

1. This is an important paper in terms of ensuring that data in a globally accessed MLST database for GBS remains clean. It is important for the scientific community to be aware of this information.
2. From the paper I assume that the tktX is only picked up through genomic based MLST analyses.
3. Recombination is relatively common in many streptococcal species. Can the authors speculate on whether the presence of TktX is due to lateral gene transfer, or a gene duplication event?
4. Is there any evidence that the strains containing tktX are genomically related? Or is it likely that the tktX was transferred multiple times to new hosts?
5. Line 124. There are two commas and 'we say' should be 'we saw'.

Response to reviewers for:

Multilocus sequence typing database for *Streptococcus agalactiae* contains a spurious allele of the transketolase gene

Each of the reviewer comments is reproduced below (in black text). Our responses follow in blue italicized text. All line numbers refer to those in the revised version of the manuscript.

Reviewer comments:

Reviewer #1:

In this article, the authors describe a *tkt* allele (*tkt_134*) of a homologous gene (*tktX*) that is not present in all *S. agalactiae* strains and can lead to erroneous MLST assignment. This is an interesting and well written research article with clear goals, study parameters and procedures. I have just one comment to be considered by the authors for revision.

It is not clear in the methods which genomic data analysis were conducted to identify and compare the *tkt_134* and *tktX* sequences and the canonical *tkt* sequence. The authors should describe these analysis more clearly in the text.

We have added an additional methods section to describe the analysis done to create Figure 1, which is the key data showing that *tkt_134* is quite different from the rest of the *tkt* alleles, leading to the designation of a “canonical *tkt*” and “*tktX*”. This section appears at lines 276-282 in the revised manuscript.

Reviewer #2:

1. This is an important paper in terms of ensuring that data in a globally accessed MLST database for GBS remains clean. It is important for the scientific community to be aware of this information.

Thanks, we agree!

2. From the paper I assume that the *tktX* is only picked up through genomic based MLST analyses.

This statement is correct, and a key point. This point was indeed only implied in the original version, so we have added a statement explicitly making this point to the discussion at line 188-191, with an additional statement about its implication at lines 195-197.

3. Recombination is relatively common in many streptococcal species. Can the authors speculate on whether the presence of *TktX* is due to lateral gene transfer, or a gene duplication event?

We considered this possibility, and believe our data are indeed more consistent lateral transfer both into GBS and between GBS clones. We have added some additional discussion of the possibilities, including recombination and duplication as noted by this reviewer, to the discussion at lines 229-243.

4. Is there any evidence that the strains contain *tktX* are genomically related? Or is it likely that the *tktX* was transferred multiple times to new hosts?

This is an insightful question, and is related to the third question above from the same reviewer. Indeed, leveraging MLST as a proxy for strain relatedness, we do see *tktX* in at least three distinct clonal lineages of GBS (ST1, ST7, and ST283). From this limited subset of strains, we have no evidence that *tkt* has been transferred multiple times into *S. agalactiae*, because the *tktX* sequence and its genomic context was nearly identical in all 3. However, we did not exhaustively search for strains with *tktX* in the databases (and don't believe that exercise would alter the conclusion or action item arising from this manuscript), so we ensured that these discussion points are clearly noted to be speculative. We have added these points to the discussion at lines 237-249.

5. Line 124. There are two commas and 'we say' should be 'we saw'.

This correction has been made.

Additional erratum:

In double-checking the figures and tables, we realized we had shown a gel for SG-M803 in Figure 3 but listed SG-M805 in Table 1. The correct strain should be SG-M803, and we have updated the data in Table 1. We have double-checked all the analyses to ensure that we did not use the data for SG-M805, and found that we indeed had correctly used genomic data for SG-M803 in all cases, and none of the results or figures required any changes due to this.

For the reader's convenience, we have also indicated whether we have detected *tktX* in the strains listed in Table 1.

Re: Spectrum00537-24R1 (Multilocus sequence typing database for *Streptococcus agalactiae* contains a spurious allele of the transketolase gene)

Dear Dr. Swaine L. Chen:

Your manuscript has been accepted, and I am forwarding it to the ASM production staff for publication. Your paper will first be checked to make sure all elements meet the technical requirements. ASM staff will contact you if anything needs to be revised before copyediting and production can begin. Otherwise, you will be notified when your proofs are ready to be viewed.

Sincerely,
Sadjia Bekal
Editor
Microbiology Spectrum

Reviewer #1 (Comments for the Author):

Dear authors,

Thanks for the answer provided and for adding more information regarding the genomic data analysis in the methods section.